# Metabolome Shift in Both Metastatic Breast Cancer Cells and Astrocytes Which May Contribute to the Tumor Microenvironment

**DOI:** 10.3390/ijms22147430

**Published:** 2021-07-11

**Authors:** Hiromi Sato, Ayaka Shimizu, Toya Okawa, Miaki Uzu, Momoko Goto, Akihiro Hisaka

**Affiliations:** 1Laboratory of Clinical Pharmacology and Pharmacometrics, Graduate School of Pharmaceutical Sciences, Chiba University, 1-8-1 Inohana, Chuo-ku, Chiba-shi, Chiba 260-8675, Japan; aykmlkty24@gmail.com (A.S.); afna6651@chiba-u.jp (T.O.); miaki.uzu@affrc.go.jp (M.U.); peachko_27.vb_j@icloud.com (M.G.); hisaka@chiba-u.jp (A.H.); 2Institute of Agrobiological Sciences, National Agriculture and Food Research Organization, 1-2 Owashi, Tsukuba, Ibaraki 305-8634, Japan

**Keywords:** metabolome shift, astrocyte, metastatic brain tumor, glutamate

## Abstract

The role of astrocytes in the periphery of metastatic brain tumors is unclear. Since astrocytes regulate central nervous metabolism, we hypothesized that changes in astrocytes induced by contact with cancer cells would appear in the metabolome of both cells and contribute to malignant transformation. Coculture of astrocytes with breast cancer cell supernatants altered glutamate (Glu)-centered arginine–proline metabolism. Similarly, the metabolome of cancer cells was also altered by astrocyte culture supernatants, and the changes were further amplified in astrocytes exposed to Glu. Inhibition of Glu uptake in astrocytes reduces the variability in cancer cells. Principal component analysis of the cancer cells revealed that all these changes were in the first principal component (PC1) axis, where the responsible metabolites were involved in the metabolism of the arginine–proline, pyrimidine, and pentose phosphate pathways. The contribution of these changes to the tumor microenvironment needs to be further pursued.

## 1. Introduction

More than 10% of patients with cancer develop metastatic brain tumors, and advanced cancers such as lung cancer and breast cancer metastasize to the brain more frequently and rapidly, making surgical resection challenging and chemotherapy refractory in many cases [1]. In recent years, it has been reported that astrocytes, a type of glial cell, contribute to the malignant transformation of cancer cells [2,3]. The blood–brain barrier, comprising vascular endothelial cells, pericytes, and astrocytes, prevents the entry of harmful substances into the brain [2,4]. In addition, astrocytes express a variety of transporters that take up and secrete numerous substances to maintain central nervous system (CNS) homeostasis, playing a role in supplying nutrients and protecting neurons [5,6].

To clarify the nature of cancer cells of the brain metastases, the interaction between cancer cells and astrocytes has been actively investigated by using their coculture systems. It has been reported that astrocytes contribute to cancer cell survival by suppressing the expression of the tumor suppressor phosphatase and tensin homolog deleted from chromosome 10 (PTEN) [2] and through endothelin signaling [7]. There are two possible pathways for the exchange of substances between cells: one is through paracrine factors such as cytokines and exosomes (including small ribonucleic acids (RNAs), such as micro RNAs (miRNAs)), and the other is through intercellular junctions such as gap junctions (GJs). GJs are formed by the aggregation of six connexin (Cx) proteins into a hexamer (hemichannel), which further aggregates between neighboring cells to form a pathway between the cytoplasm of both cells. In particular, connexin 43 (Cx43) is abundantly expressed in astrocytes, and the network constructed by a GJ has been found to affect neural activity in the CNS [8,9]. Therefore, if cancer cells infiltrating the brain utilize astrocyte functions for their survival, a GJ may be an intervening molecule in this process [10].

However, in the early stages of metastasis, astrocytes have also been reported to contribute to cancer suppression by inducing cell death via the secretion of plasminogen activator to produce plasmin, which suppresses brain metastasis by converting membrane-bound astrocytic Fas ligand into a paracrine death signal for cancer cells [11]. Astrocytes also interfere with the metastasis-promoting signal introduced by the downstream signal of the transient receptor potential ankyrin 1 protein (TRPA1), an ion channel, via miRNA (miRNA-142-3p) secretion [12]. Taken together, these results indicate that the state of the brain microenvironment, especially that of the astrocytes, comprising most of the brain composition, changes dynamically during sustained contact with cancer cells.

It is generally believed that cancer cells acquire an abnormal proliferative capacity by activating glucose and amino acid metabolism [13]. On the other hand, astrocytes are regulators of the CNS metabolism, and the metabolites they produce, secrete, and recover affect the survival of the surrounding cells. In other words, a certain relationship between metabolic changes and cellular phenotypes, such as proliferation and motility, is expected, and metabolome shift can be considered as a consequence of early interactions that can occur in the microenvironment. We hypothesized that astrocytes are metabolically altered by cancer cells, resulting in changes in their physiological functions and that direct or indirect contact with altered astrocytes affects the malignant phenotypes of cancer cells. To test this hypothesis, we cocultured astrocytes and cancer cells and aimed to clarify what metabolome shift occurred in both cells, as well as the phenotype transition. In metabolomics, it is possible to capture the metabolic flow of the entire cell by comprehensively measuring small molecules such as sugars, amino acids, and nucleic acids in the cell [14]. Therefore, we measured 110 compounds using capillary electrophoresis time-of-flight mass spectrometry (CE-TOFMS) and extracted metabolic pathways containing metabolites that showed significant changes to evaluate the transformation of astrocytes and cancer cells, respectively, and to clarify the mechanism of early metastatic brain tumor progression from the viewpoint of cellular metabolism.

## 2. Results

### 2.1. Alteration of Cancer Malignant Phenotype by Coculture with Astrocytes

First, to clarify the effect of astrocytes on several cancer characteristics, we tested two types of cocultures: treatment with culture supernatant (indirect coculture) and mixed seeding (direct coculture).

The change in cisplatin (CDDP) sensitivity of MDA-MB-231 (MDA231) cells when cocultured with astrocytes based on Annexin V, an indicator of early apoptosis, was examined. CDDP concentrations were set at 60 µM and 120 µM, which are the IC_50_ values of MDA231 in single culture, and the concentration at which the cells are expected to undergo apoptosis, respectively. As a result, apoptosis was significantly inhibited at 120 µM in the direct coculture group, while there was no change in the indirect coculture group (Figure 1A,B).

The migration ability of MDA231 cells when astrocytes or NIH3T3 cells were seeded on the well side as attractants was then evaluated by the number of MDA231 cells permeabilizing the chamber insert to the opposite side of the membrane. In the case of astrocytes in the wells, the number of MDA231 cells permeating the membrane was significantly higher than that in the control (no cells in the bottom well); however, no significant difference was observed in the case of NIH3T3 cells compared with the control, confirming that the increase in the migration ability of MDA231 cells was specific to astrocytes (Figure 1C,D).

Next, the effect on the proliferation of cancer cells was examined. In the mixed seeding coculture group, astrocytes significantly inhibited the growth rate of MDA231 compared with MDA231 single culture (Figure 1E, right, direct coculture). This trend was not observed in the group treated with astrocyte culture supernatant (Figure 1E, left, indirect coculture). Considering the possibility that astrocytes suppress cell cycle progression in MDA231, we next examined the expression and localization of Ki-67, a well-known proliferative marker in breast cancer. It is expressed and changes its localization during mitosis; it is diffused throughout the nucleus in the G2/M phase (Figure 1F photo, positive-diffuse), and gathered locally in the G1 phase of the next cell cycle after the M phase (Figure 1F photo, positive-local). This property can be used to predict the cell cycle progression (Figure 1F photo, illustration) [15]. Contrary to expectations, there was no significant change in Ki-67 localization in the astrocyte coculture group, and no Ki-67-negative cells were observed. On the other hand, when paclitaxel, a mitotic inhibitor, was added to promote M-phase arrest, positive-diffuse was greatly reduced and Ki-67-negative cells appeared in the group cocultured with astrocytes. These results suggest that the presence of astrocytes may help cancer cells escape from the M-phase arrest and enter the interphase (G0) when stress is applied to the cancer (Figure 1G pie chart).

### 2.2. Metabolome Shift of Astrocytes by Treatment with Culture Supernatant of MDA231

To investigate the interaction between astrocytes and MDA231 from the viewpoint of intracellular metabolism, cells were treated with culture supernatant of each other (indirect coculture) and the changes were evaluated by metabolome analysis. First, to examine the changes on the astrocyte side, intracellular metabolites were collected 48 h after treatment with MDA231 culture medium, and then indirect coculture and single cultured astrocytes were compared. Astrocytes exhibit quiet properties, for example, they are less responsive to cytotoxic agents and grow more slowly than cancer cells. Based on this characteristic, a time point was set that would allow for steady changes. A period of 48 h was reasonable in this regard. Seven metabolites, spermidine, adenosine, homoserine, glycine, lysine, ornithine, and guanosine, significantly increased more than two-fold and glyoxylic acid was significantly decreased by less than half in the indirect coculture group (Figure 2A, Appendix A). Pathway analysis was performed on the eight metabolites, and the metabolic pathways are shown in descending order of “(−log (Raw *p*))×Impact” in the following figures to take into account both the *p*-value and impact, which is an indicator of the influence on other metabolic pathways. In particular, amino acids such as arginine, ornithine, and proline showed an increasing trend in the coculture group (Figure 2C, Appendix A). In this study, we focused on glutamate (Glu) among the metabolites in this pathway because of its high centrality score in the pathway (Appendix A) and its importance in the CNS metabolism by astrocytes.

### 2.3. Effects of Culture Supernatant of Glu-Exposed Astrocytes (GluAM) on Proliferation and Migration of MDA231

To investigate the effect of the metabolomic shift of astrocytes by coculture with cancer cells on the proliferation and migration of MDA231 cells, we prepared Glu-exposed astrocytes as a pseudo-condition for altered astrocytes according to the results of Section 2.2. As a control, we also set a group wherein Glu was directly added to the cancer cell media. Cell viability was checked after 8 h and 48 h of treatment with astrocyte culture supernatant (indirect coculture), and fold change of the viability was determined by dividing cell viability at 48 h by that at 8 h. As a result, Glu had no direct effect on cell proliferation (Figure 3A). In addition, there was no significant change in the group containing 50% astrocyte culture supernatant (50%AM), in the group using culture supernatant of Glu-exposed astrocytes (GluAM), or in the group containing threo-β-benzyloxyaspartate (TBOA), which inhibits Glu uptake transporter, excitatory amino acid transporter (EAAT) (GluTBOAAM), whereas the cell viability of GluAM tended to increase slightly at 48 h (Figure 3A). The effect of the same treatment on the migration of MDA231 was also examined. When the astrocyte culture supernatant was placed in the well, there was almost no change (Figure 3B), while the number of migrating cells increased significantly when astrocytes were placed in the well (Figure 3C). Neither Glu treatment nor inhibition of Glu uptake in astrocytes directly affected the migration phenotype.

Although the Glu exposure conditions, which are assumed to be the initial metabolic transformation in astrocytes, do not seem to have a direct effect on the malignant phenotype of MDA231, we next investigated the metabolomic changes of MDA231 as a comprehensive indicator of the state changes immediately below the apparent phenotype.

### 2.4. Metabolome Shift of MDA231 by Treatment with Astrocyte Culture Supernatants

Astrocytes were treated with or without Glu exposure, as described in Section 2.3, (Figure 4A), and MDA231 cells were cocultured with the culture supernatant (indirect coculture) to determine whether there was any change in intracellular metabolism. When only Glu was added directly to the medium of cancer cells (group 2, MDA231+Glu), there was almost no change in metabolites compared with the control (group 1, MDA231) (Figure 4B,C, Appendix A). In contrast, when Glu-treated astrocyte culture supernatant was added to the medium of cancer cells (group 4, MDA231 + Astro Medium exposed to Glu), significant changes in many metabolites were observed (Figure 4B,C). In particular, the changed metabolites clustered from deoxyadenosine triphosphate to cytidine monophosphate (squares in Figure 4B) were very similar to those observed when only astrocyte culture supernatant was added (group 3, MDA231 + Astro Medium) and disappeared to the same extent as in the control when TBOA was added (group 5). These compounds overlapped by 75% with the metabolites that explained 50% of the first principal component (PC1) in the principal component analysis (PCA) (Appendix A). To avoid overlooking small changes, we selected metabolites that explained 50% of PC1 as a selection criterion for pathway analysis. These pathways include arginine and proline metabolism, which are also highly altered in astrocytes; the pentose phosphate pathway, which is involved in energy metabolism; and pyrimidine metabolism, which is involved in nucleic acid metabolism (Figure 4D).

To further investigate the effect of astrocytes exposed to Glu on the metabolic shift of MDA231 cells in detail, a comparison between each of the two groups was made, and the metabolites extracted by volcano plots were used for pathway analysis (Appendix A). The results showed that astrocyte culture supernatant and MDA231 (1 vs. 3), Glu-exposed astrocyte culture supernatant and MDA231 (1 vs. 4), astrocyte culture supernatant and Glu-exposed astrocyte culture supernatant (3 vs. 4), and Glu-exposed astrocyte culture supernatant and inhibition of Glu uptake by TBOA (4 vs. 5), all showed the common pathways of large changes (Figure 4E). Considering the results shown in Figure 4D, glutamate-associated metabolism, the pentose phosphate pathway, and pyrimidine metabolism are involved in the entire interaction. It was inferred that the enhancement of the tricarboxylic acid (TCA) cycle also played a role in the interaction with astrocytes, which became more reactive after Glu exposure.

## 3. Discussion

We evaluated two types of culture systems: direct coculture (direct contact between astrocytes and cancer cells) and indirect coculture (contact via liquid factors, i.e., paracrine factors). The important mechanism in direct contact is the existence of gap junction (GJ). However, from the viewpoint of interacting molecules, paracrine factors that are secreted outside the cell and then taken up by the other cell or stimulate the receptor are also evaluated in this direct coculture system. If the contribution of GJ-impermeable paracrine factors to the phenotype under evaluation (drug sensitivity, proliferation, migration, etc.) is not large, no difference can be confirmed by indirect coculture, but direct coculture includes the influence of paracrine factors in addition to GJ-permeable factors. It is possible that the GJ-permeable factor and the paracrine factor are common (i.e., there is a transduction pathway other than GJ). On the other hand, if a difference is found in our indirect coculture, it is clear that the interaction occurred in a GJ-independent transduction pathway.

In the present study, direct coculture showed a decrease in drug sensitivity of MDA231 (Figure 1B) and a tendency to inhibit growth (Figure 1E). Indirect coculture using culture supernatant collected at a fixed point did not show a clear change in phenotypes. However, the presence of astrocytes significantly promoted the migration of MDA231 by the indirect coculture (Figure 1D). It was the only one that could be tested in which both cells remain communication separated by a chamber. Hohensee et al. reported that cancer cells in the transwell insert in the presence of astrocytes under the insert showed increased migration ability [16], and Kim et al. reported that cancer cells were resistant to various anticancer drugs by direct coculture with astrocyte [17]. Our results support these reports. It is possible that by prolonging the reaction time or increasing the intensity of exposure, the influence may be seen even with indirect coculture; however, it is thought that direct coculture strengthens the cancer phenotype more aggressively.

It is also suggested that factors that contribute significantly to cancer cell migration and survival or proliferation are different. The intervening molecules that exert their effects on migration may be those that are released and taken up extracellularly and have multiple pathways for their intracellular and extracellular transport, or those that act on surface receptors to promote downstream signaling. Hohensee et al., who showed that PTEN loss has important implications for the metastatic properties of triple-negative breast cancer (TNBC), found several cytokines released into the supernatant when cocultured with astrocytes, particularly in the context of their close involvement with PTEN, and identified granulocyte-macrophage-colony stimulating factor (GM-CSF) (CSF2) as one of them [16]. Although they could not distinguish between TNBC and astrocytes as the source of GM-CSF release in their experimental system, there was a high correlation between GM-CSF expression in TNBC tissues and decreased survival [18]. In addition, the CSF2 receptor appeared on astrocytes only in cocultures with tumor cells [16], suggesting that GM-CSF is released from TNBCs and astrocytes respond by releasing some chemotactic substances, which may contribute to enhanced migration.

On the other hand, a direct coculture may be influenced by GJs formed by metastatic brain tumors and surrounding astrocytes. The protective effects of astrocytes against drug-induced apoptosis of cancer cells include the sequestration of calcium directly related to cell death from the cytoplasm of tumor cells via GJs, and additional mechanisms have been suggested, including upregulation of survival genes such as BCL2L1, TWIST1, and GSTA5 in tumor cells [11]. It has also been confirmed that 2′3′ cyclic GMP-AMP (cGAMP) moves through GJs formed between cancer cells and astrocytes, and accelerates the stimulator of interferon genes (STING) pathway in astrocytes, which finally promotes cytokine secretion such as tumor necrosis factor (TNF), which in turn assists cancer cell survival [3]. Importantly, the loss of GJs by Cx43 knockdown in breast cancer cells did not suppress brain metastasis; however, it suppressed tumor colony growth in the brain. In addition to cGAMP, it has been reported that miR-709, which is involved in lung cancer progression, may permeate GJs in a coculture system of metastatic lung cancer PC14 and astrocytes [19]. Substances that permeate GJs may directly affect cell survival and subsequent growth. In this study, we did not directly identify any changes in the expression of antiapoptotic or proapoptotic factors that might explain the interaction, which needs to be pursued in the future.

Taken together, the present study clearly shows that it is likely that GJ-dependent mechanisms are strongly involved in drug sensitivity and cell proliferation, while GJ-independent transduction pathways are involved in the interaction for migration. However, since GJ-permeable substances and paracrine factors may be common, it is difficult at present to narrow down the interacting molecules that affect each phenotype or to estimate their influence. Furthermore, another challenge in the mixed seeding conditions is the need to carefully compensate for the effect of metabolite loss due to cell separating manipulation by cell sorting on metabolome evaluation [20]. Considering the technical challenges, in proving our hypothesis that the interaction between cancer cells and astrocytes would first appear in each other’s metabolome, we applied the evaluation system of indirect coculture to metabolome analysis, expecting the contribution of paracrine factors not limited to GJ permeabilization.

First, the changes in astrocytes induced by MDA231 were particularly pronounced in amino acid metabolism. Astrocytes are known to take up Glu produced by neural activity to suppress the cytotoxicity caused by excess Glu [8]. Considering that the metabolic pathways of Glu-related substances such as arginine and proline were extracted from pathway analysis when cancer cell supernatants were added to astrocytes (Figure 2B,C), the interaction between cancer cells and astrocytes is likely to depend on the major physiological function of astrocytic Glu uptake and associated Glu-peripheral metabolism. Based on this hypothesis, we attempted to culture MDA231 cells with astrocyte culture supernatants that were exposed to Glu, to evaluate the effect on the phenotype and metabolism of cancer cells.

The metabolites with high contribution of PC1 in PCA showed an increasing trend with astrocyte culture supernatant (group 3) compared to control (group 1). It also occurred both when only astrocyte culture supernatant was added (group 3) and when astrocyte supernatant cultured under Glu-exposed conditions was added (group 4) (Figure 4C,E). This supports the hypothesis that the metabolic changes, which occurred in cancer cells when astrocyte supernatants were added, depended on “Glu uptake and associated peri-Glu metabolism by astrocytes.” Since the addition of Glu alone did not cause significant changes in the MDA231 metabolome (Figure 4B,C), Glu itself was not directly responsible for the metabolic changes.

Pathway analysis showed that the arginine–proline pathway, which promotes polyamine synthesis and is involved in RNA translation and cell proliferation, contributes to changes in both MDA231 and astrocytes. On the other hand, it was observed that mixed coculture with astrocytes tended to decrease the proliferation rate of MDA231 (Figure 1E) as described above. In the early stage of brain metastasis, when the survival of cancer cells is unstable, it has been reported that cancer cells slow down cell cycle progression and enhance apoptosis-resistant survival signals. This switching is thought to involve a mechanism in which a component secreted by peripheral astrocytes suppresses DNMT1, a methyltransferase of cancer cells [21]. It is suggested that there is a mechanism by which the responsiveness of peripheral astrocytes is dynamically altered upon some stimuli or oxidative stress loading and that such changes are first manifested in metabolomic shifts. We have previously shown that the histone deacetylase inhibitor, trichostatin A, attenuates drug resistance by metabolic shift in renal cell carcinoma [22], and epigenetic regulation may consequently induce a metabolomic shift [23,24]. Considering the suppressive effect against proliferation in the coculture group, the arginine–proline pathway in this context may involve the promotion of certain gene expression rather than cell growth.

The pentose phosphate pathway (PPP) is a major source of nicotinamide adenine dinucleotide phosphate (NADPH) and enhances metabolic flux [25], which may lead to cell growth. The TCA cycle is generally incomplete and slows down in cancer cells [26]; however, mitochondrial oxidative phosphorylation (OXPHOS) is often maintained [27,28], allowing large amounts of ATP production, which is effective for energy production throughout the entire cell. However, the activation of OXPHOS also promotes the production of reactive oxygen species (ROS) from oxidative stress sources, and the capture of ROS by NADPH is important for cell survival. In addition, the TCA cycle produces NADH and FADH_2_, which serve as important electron carriers for OXPHOS. Therefore, activation of both the PPP and TCA cycle may be beneficial for cancer cell survival. Although cancer stem cells (CSCs) are known for their glycolytic-dependent metabolism and high glucose uptake capacity, certain quiescent or low-metabolism-turnover tumor-initiating CSCs, including breast cancer and glioblastoma, are more dependent on OXPHOS than differentiated progenitor cancer cells [28,29,30]. Conversely, when OXPHOS is inhibited, there is a shift to glycolytic-dependent metabolism [30], and trends in metabolomic changes suggest that the cancer microenvironment, where astrocytes are the majority, contributes to this early CSC flexibility.

Although we were unable to fully examine the metabolome under the direct coculture conditions, it was preliminarily observed that direct coculture of MDA231 and astrocytes resulted in a greater degree of change compared with the indirect coculture performed in this study (data not shown). While GJs have been identified between astrocytes and cancer cells by mixed seeding [3,19,31], the causal relationship between this presence and metabolomic changes in both cells is unknown. However, changes in cellular phenotype due to GJ-mediated exchange of molecules may allow for rapid response independent of gene expression regulation [32], which may be an advantage for cancer cells with survival at stake. For dynamic astrocyte reactivity in response to circumstances, molecules transferred from cancer cells in contact via GJs would be a direct trigger for a metabolomic shift. It is also possible that the molecule is common with a paracrine factor that is secreted out of the cell and taken up into the other cell in sustained cultures of both cells. In this study, we proposed the hypothesis that there are intracellular metabolomic changes prior to the appearance of cancer malignant phenotype on the surface, and actually identified metabolic pathways that were altered in conjunction with contact. The significance of these enhanced metabolic pathways found in the present study needs to be explored by assessing the impact of these changes on the malignant phenotype of cancer under experimental conditions in which cell–cell interactions are maintained.

## 4. Materials and Methods

### 4.1. Reagents

All cultures and reagents were procured from Sigma Chemical Company (St. Louis, MO, USA) unless otherwise indicated. Cellmatrix Type IC (Nitta Gelatin Inc., Osaka, Japan) was diluted with HCl (pH 3.0) to make 0.1 mg/mL type-I collagen solution. CDDP (Wako Pure Chemicals, Osaka, Japan) was prepared in dimethyl sulfoxide (DMSO) (Wako Pure Chemicals, Osaka, Japan) at a concentration of 100 mM just before use, and the final concentration of DMSO in the cell culture was <0.1%. DL-TBOA (R&D Systems, Minneapolis, MN, USA) was used as an inhibitor of the glutamate uptake transporter (EAAT2). Precrystalline deoxyribonuclease (DNase) I (Wako Pure Chemicals), from bovine pancreas, was dissolved in distilled water to make a 25 mM stock solution.

### 4.2. Cell Culture

Astrocyte cultures were prepared from neonatal Wistar rats as described previously by Kajitani et al. [33]. Briefly, the isolated cerebral cortices and hippocampi were minced and incubated with trypsin and DNase. Dissociated cells were plated in 75 cm^2^ tissue culture flasks (8–15 × 10^6^ cells/flask) precoated with 0.1 mg/mL type-I collagen solution. After 8–12 days, the cells were purified to remove fewer adherent neurons and microglia with a rotary shaker at 100 rpm for 15 h. Adherent cells were trypsinized (0.25%) and plated into 75 cm^2^ flasks. After the cells reached confluence (10 days), the confluent cells were shaken by hand for 10 min. Adherent cells were trypsinized (0.25%) and plated on new dishes. Using this method, more than 90% of the cells expressed glial fibrillary acidic protein (GFAP), a marker of astrocytes. Animal experiments were performed in accordance with the protocols approved by the Animal Research Committee of Chiba University.

The human breast cancer cell line MDA231 was obtained from the American Type Culture Collection (Manassas, VA, USA). MDA231 cells were transfected with the enhanced green fluorescent protein (eGFP) using the pEGFP-N1 plasmid with Lipofectamine 3000 (Invitrogen, Carlsbad, CA, USA) according to the manufacturer’s instructions. To obtain stable transfected cells, cells were further subjected to cell sorting using BD FACS AriaTM II (BD Biosciences, San Jose, CA, USA). The pEGFP-N1 plasmid was gifted by Dr. Aoki, S. to Chiba University.

The murine fibroblast cell line NIH3T3 was used as a control for astrocytes. NIH3T3 was gifted by Dr. Nishimura, K. to International University of Health and Welfare.

All cells were grown in Dulbecco’s modified Eagle’s medium (DMEM) (Wako Pure Chemicals) supplemented with 10% fetal bovine serum (FBS) (Biowest, Riverside, MO, USA), penicillin (0.5 U/mL), and streptomycin (1 µg/mL), and were maintained at 37 °C in a fully humidified atmosphere of 5% CO_2_.

### 4.3. Coculture of Astrocytes and MDA231

Two types of coculture methods were used: (1) indirect coculture: addition of culture supernatant (generally used a 3-day culture medium of astrocytes or MDA231 diluted to 50% with DMEM was used), and (2) direct coculture: mixed seeding (astrocyte cell number:MDA231 cell number = 1:1), except for the migration assay, where cells were incubated with the sharing medium during the coculture period as MDA231 in a cell-culture insert with 8.0 µm pores and astrocytes in a well of 24-well plate under the insert.

### 4.4. Migration Assay

The migration assay was performed as previously described [34]. Briefly, MDA231 was treated with FBS-free DMEM for 24 h before the day of the experiment. As an attractant of migration, a total of 500 µL DMEM with 10% FBS containing 5 × 10^4^ astrocytes or NIH3T3 cells was seeded in a well of a 24-well plate, and then MDA231 cells were trypsinized and suspended in an FBS-free medium. A total of 200 µL of FBS-free medium containing 5 × 10^4^ cells was placed into the top of the migration chambers with 8 µm filters (24-well plate format; Becton Dickinson Labware, Franklin Lakes, NJ, USA), which were placed in wells containing astrocytes or NIH3T3 cells. The cells were incubated at 37 °C and 5% CO_2_ for 4 h or 8 h. Following this, the chambers were then removed from the wells and coded for analysis by a blinded observer. Cells that had migrated to the bottom of filters were fixed with methanol, rinsed thoroughly with phosphate-buffered saline (PBS), and stained with Giemsa’s solution. Four visual fields were randomly selected for each chamber, and the total number of cells at the bottom of each field was counted under a microscope and then averaged between the four.

### 4.5. Cell Proliferation Assay

MDA231 was cocultured with type (1) or (2) astrocytes as described in Section 4.3. The eGFP fluorescence intensity of MDA231 was measured at 488 nm excitation and 515 nm emission using SpectraMax^TM^ i3 (Molecular Devices, Sunnyvale, CA, USA), 24, 48, and 72 h after the start of coculture. It was confirmed that eGFP emission is proportional to cell number in advance (Appendix A).

### 4.6. Apoptosis Assay

Cells were arranged at a density of 2.0 × 10^5^ in a 60 mm dish for one tube of flow cytometry. MDA231 single culture or MDA231 cells with astrocyte culture supernatant (50% conditioned medium diluted by DMEM) were seeded at 2.0 × 10^5^ cells/well in a 6-well plate. In the mixed seeding of MDA231 and the astrocyte group, cells were mixed in a 1:1 ratio to obtain a total of 4 × 10^5^ cells/well. After 48 h of culture, the medium was removed and replaced with 120 µM CDDP medium. After an additional 48 h, the cells were trypsinized and washed with PBS. Cells were resuspended in Annexin V binding buffer (BioLegend, San Diego, CA, USA) to 1 × 10^6^ cells/mL. Cell fluid (100 µL) was taken and 5 µL of allophycocyanin (APC) -Annexin V (BioLegend) and 10 µL of propidium iodide (PI) (Invitrogen, Carlsbad, CA, USA) were added, mixed gently, and incubated for another 15 min in the dark. Finally, 400 µL of Annexin V binding buffer was added. The cell fluid was passed through a mesh and transferred to a fluorescence-activated cell sorting tube for flow cytometry using BD FACS Canto II^TM^ (BD Biosciences). For the early apoptotic marker, the Annexin V-positive population was detected, while the late apoptotic marker, PI-positive population, was detected. Finally, Annexin V-positive and PI-negative populations in each culture group were compared.

### 4.7. Immunofluorescence Staining

A total of 4 × 10^5^ cells were seeded on coverslips, which were placed in a 12-well plate for 24 h. The cells were treated with 20 nM paclitaxel or DMSO (control) for 24 h and then fixed in 4% paraformaldehyde PBS (Wako Pure Chemicals, Tokyo, Japan) for 20 min at 37 °C. After blocking with PBS containing 0.1% saponin (MP Biomedicals, Santa Ana, CA, USA) and 3% bovine serum albumin (BSA) for 30 min, the cells were incubated with primary antibodies for 1 h, followed by incubation with secondary donkey anti-rabbit IgG H&L, Alexa Fluor 405 (Abcam, Cambridge, UK, ab175651) or goat anti-mouse IgG (H+L), Alexa Fluor 647 (Thermo Fisher Scientific, Boston, MA, USA)-conjugated antibodies (diluted in PBS containing 0.1% saponin, 1:500) for 1 h; all reactions were carried out at room temperature. Coverslips were picked and mounted with DAPI-Fluormount-G^®^ (SouthernBiotech, Birmingham, AL, USA). Finally, the cells were observed under a Zeiss LSM 780 confocal microscope (Carl Zeiss MicroImaging GmbH, Jena, Germany). The dilutions used for the primary antibodies were as follows: GFAP (1:300; Bioss, Woburn, MA, USA, bs 0199R) and Ki67 (1:200) (Proteintech, Rosemont, IL, USA, 27309-1-AP). All primary antibodies were diluted in PBS containing 0.1% saponin and 3% BSA. The obtained images were analyzed using the ImageJ2 software (National Institute of Health, Bethesda, MD, USA).

### 4.8. Metabolite Extraction and Metabolome Analysis Using CE-TOFMS

Extractions were performed according to a previous report by Hatakeyama et al. [35]. Briefly, 1.0 × 10^6^ astrocytes or MDA231 were collected in one tube. Cells were detached by trypsin-EDTA treatment and washed twice with 5% mannitol. Cell pellets were suspended in 1 mL of methanol, sonicated for 30 s, and vortexed with 1 mL of chloroform and 0.4 mL of ultra-pure water for 30 s. After centrifugation at 2300× *g* at 4 °C for 5 min, the aqueous layers were filtered through a Nanosep/3K (3-kDa cut-off) filter (Nihon Pall Ltd., Tokyo, Japan) at 9100× *g* at 4 °C for 2.5 h to remove proteins and phospholipids. The resultant filtrates were used as ionic metabolites; they were lyophilized and dissolved in 25 μL ultra-pure water prior to CE-TOFMS analysis. CE-TOFMS was carried out using an Agilent 7100 CE system equipped with an Agilent 6230 TOF-MS system (Agilent Technologies, Santa Clara, CA, USA). Raw data were processed using the Mass Hunter software (Qualitative and Quantitative Analysis, Agilent Technologies, Santa Clara, CA, USA) for metabolite quantification.

### 4.9. Statistical Analysis

Each experiment was performed independently at least three times and data are presented as mean ± standard deviation (SD) or standard error of the mean (SEM), except for metabolome analysis. Statistical analysis was performed using the Tukey–Kramer test for comparing multiple groups, or t-test for comparing two groups. *p*-values < 0.05 were considered statistically significant. Regarding metabolome analysis, for clustering analysis (heat map), each square indicates a normalized value derived from three samples in each group. Analysis was performed by statistical analysis and pathway analysis using MetaboAnalyst 4.0 (http://www.metaboanalyst.ca/ (accessed on 10 May 2021)). In statistical analysis, the number of metabolites was converted into a Z score and analyzed. The fold changes and *p*-values for the metabolites in the astrocyte and MDA231 coculture cells as compared with astrocyte or MDA231 single culture cells were calculated. For the volcano plot, the *p*-values were transformed by −log10 so that the more significant features (*p*-values < 0.1 as statistically significant) could be expressed higher on the graph.

### 4.10. Pathway Analysis

This study elucidated aspects of both the metabolic enrichment of pathways using metabolite set enrichment analysis and the biological importance of the metabolic pathway detected by the centrality theory via topology analysis. The differentially changed metabolites extracted from the volcano plot metabolome or heat map were subjected to pathway analysis with the MetaboAnalyst 4.0 platform (http://www.metaboanalyst.ca/ (accessed on 10 May 2021)), according to previous reports [22,36,37].

## 5. Conclusions

In this study, we confirmed the possibility that metastatic brain tumors use and interact with astrocytes as a basis for survival in the early stages of metastasis via metabolic alteration. The metabolome changes in both cells are expected to precede changes in the cell phenotype that become apparent. It is essential to investigate the effects of the alterations in the arginine–proline pathway observed in astrocytes, and in the arginine–proline pathway, pyrimidine metabolism, pentose phosphate pathway, and TCA cycle observed in cancer cells, as well as to identify and elucidate the mechanisms of the paracrine factors that cause metabolomic changes.

## Figures and Tables

**Figure 1 ijms-22-07430-f001:**
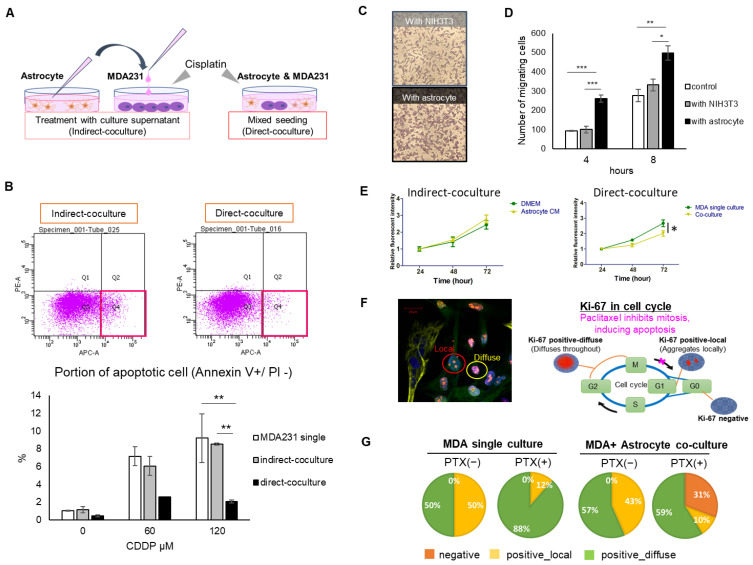
Effect of astrocyte coculture on MDA231 malignant phenotype. (**A**) Two types of cocultures were attempted: treatment with astrocyte culture supernatant (indirect coculture) and mixed seeding (direct coculture). Apoptosis introduced by cisplatin into MDA231 was examined by flow cytometry after coculture. (**B**) Quantitative results of the population of early apoptosis (Annexin V-positive, PI-negative) in CDDP 120 µM group, represented by the position of the red square in A. Shows PE-A, propidium iodide (PI) staining area, late apoptosis markers. Shows APC-A, Annexin V staining area, early apoptosis markers. Data are expressed as mean ± SD (n = 3, except for 60 µM direct coculture; n = 2). ** *p* < 0.01, Tukey–Kramer test. (**C**) Cells were migrated via a Boyden chamber insert with NIH3T3 or astrocytes in the lower well. (**D**) The average number of cells migrated in the observed field. Data are expressed as mean ± SEM (n = 3). * *p* < 0.05, ** *p* < 0.01, *** *p* < 0.001, Tukey–Kramer test. (**E**) Cell viability measured by the fluorescence intensity of enhanced green fluorescent protein (eGFP) cocultured with astrocyte culture supernatant (indirect coculture) or the astrocyte itself (direct coculture). Data are expressed as mean ± SEM (n = 3). * *p* < 0.05, Student’s t-test. (**F**) Photo: Ki-67 staining locally or globally diffused in the nucleus. Illustration: Relationship between Ki-67 expression and localization in the cell cycle and mitotic processes. (**G**) Pie chart of the effect of astrocytes on Ki-67 status in MDA231 with or without paclitaxel (PTX) treatment. Data show the average value of the four fields in each group.

**Figure 2 ijms-22-07430-f002:**
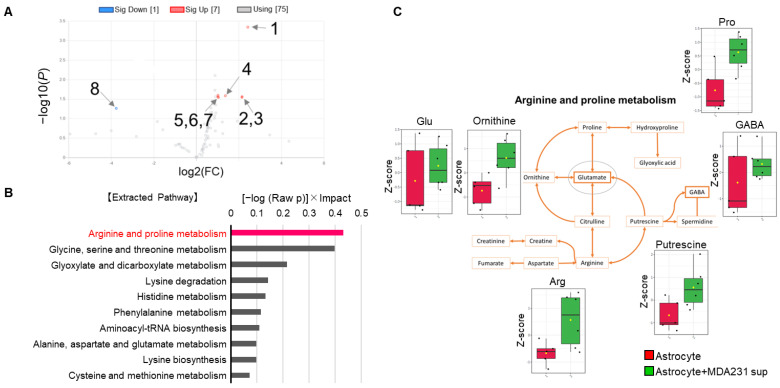
Metabolome changes in astrocytes by indirect coculture with MDA231. (**A**) Volcano plot of altered metabolites after indirect coculture (n = 6) for a single cultured astrocyte (n = 4). 1—Spermidine; 2—Adenosine; 3—Guanosine; 4—Glycine; 5—Ornithine; 6—Homoserine; 7—Lysine; 8—Glyoxylic acid. See the supplementary Appendix A. (**B**) Pathways extracted by metabolome enrichment analysis from metabolites selected by volcano plots. The bar length indicates the value of “(−log (Raw *p*))×Impact” in descending order. (**C**) Arginine and proline metabolic pathways with the detected metabolites (n = 3) as a Z-score. Data scaling (Z-scoring) was performed by MetaboAnalyst 4.0 as mean-centered and divided by standard deviation of each variable.

**Figure 3 ijms-22-07430-f003:**
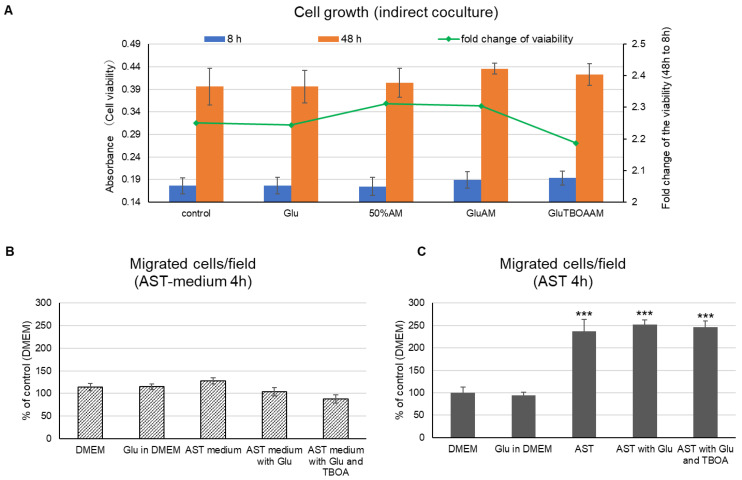
Effect of glutamate (Glu) exposure to astrocytes on the MDA231 phenotype. (**A**) Cell growth of MDA231 after indirect coculture. Control—MDA231 cultured in Dulbecco’s modified Eagle’s medium (DMEM); Glu—MDA231 cultured in 3 µM Glu; 50% astrocyte culture supernatant (50%AM) —MDA231 performed indirect coculture with 50% diluted astrocytes culture supernatant; GluAM—MDA231 performed indirect coculture with astrocytes exposed to 3 µM Glu; GluTBOAAM—MDA231 performed indirect coculture with astrocytes exposed to 3 µM Glu and 100 µM of threo-β-benzyloxyaspartate (TBOA) (excitatory amino acid transporter (EAAT) inhibitor). Bar graphs show cell viability at 8 h or 48 h, showing mean ± SEM (n = 6). The line graph shows fold change of the viability calculated by dividing the cell viability at 48 h by that at 8 h. (**B**,**C**) Cells were migrated via the Boyden chamber insert after the same procedure as in (**A**,**B**) in the case of indirect coculture or (**C**) in the presence of astrocytes in lower well. Data are shown as mean ± SD (n = 3). *** *p* < 0.001, Tukey-Kramer test vs. DMEM (control) group.

**Figure 4 ijms-22-07430-f004:**
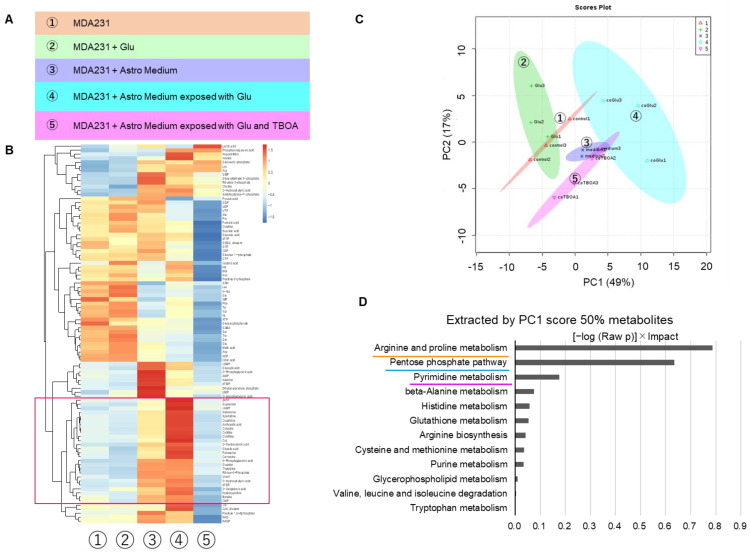
Metabolome changes in MDA231 by indirect coculture with astrocytes. (**A**) Treatment group was shown. 1—MDA231 cultured in DMEM; 2—MDA231 cultured in 3 µM Glu; 3—MDA231 performed indirect coculture with 50% diluted astrocytes culture supernatant; 4—MDA231 performed indirect coculture with astrocytes exposed to 3 µM Glu; 5—MDA231 performed indirect coculture with astrocytes exposed to 3 µM Glu and 100 µM of TBOA (EAAT inhibitor). (**B**) Heat map of clustering analysis. The red square metabolites show significant changes in groups 3, 4, and 5 in common compared with group 1. (**C**) 2D plot of principal component analysis (PCA). Numbers show each treatment group. (**D**) A pathway extracted by metabolome enrichment analysis derived from metabolites that contributed 50% to the first principal component (PC1) component of PCA. The bar length indicates the value of “(−log (Raw *p*))×Impact” in descending order. (**E**) 2D plot of PCA is shown with an arrow attached between each of the two groups. Comparison of each of the two groups was performed by metabolome enrichment analysis derived from metabolites selected by volcano plot. The bar length indicates the value of “(−log (Raw *p*))×Impact” in descending order.

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
