# Peer review of "Metabolome Shift in Both Metastatic Breast Cancer Cells and Astrocytes Which May Contribute to the Tumor Microenvironment"

_ijms, 2021, doi:10.3390/ijms22147430_

Round 1

Reviewer 1 Report

"Metabolome shift in both metastatic breast cancer cells and astrocytes which may contribute to the tumor microenvironment" is an interesting study exploring potential bidirectional relationships between astrocytes and MDA231, used here as model cells for metastatic brain tumors. The manuscript is well-written, easy to follow, provides extensive references and discussion and the results support most conclusions. A few points should, however, be addressed to allow a better appreciation of the significance of these findings. 

  1. Please add a description of the type and number of samples/replicates used to calculate statistical significance, as well as whether the data are means ± SEM or means ± SD for each experiment.
  2. Please add statistical significance in fig. 3.
  3. The authors found that co-culture with astrocytes changed the phenotype of MDA231 and then performed metabolic analysis, presumably to identify key metabolic factors responsible for the phenotype change. From the analysis, they focused on glutamate (Glu) as such a factor. However, while Glu-exposed astrocytes induced metabolic changes in MDA-MB-231, they failed to reproduce the phenotype changes. How did the authors validate that Glu-exposed astrocytes can act as a "pseudo-condition for altered astrocytes"? Did they try astrocytes treated with other factors from their analysis?

Reviewer 2 Report

In my opinion the work is well written and presents some very interesting data. I report some considerations:

  • The authors examined the change in cisplatin (CDDP) sensitivity of MDA-MB-231 (MDA231) cells when cocultured with astrocytes based on Annexin V, an indicator of early apoptosis. As a result, apoptosis was significantly inhibited at 120 µM in the direct co-culture group, while there was no change in the indirect coculture group. In my opinion they have to better clarify the reason for the selected concentration and in my opinion have to add gene or protein modulation of pro or antiapoptotic factors. Moreover, this analysis at different times could be one more point to better evaluate the work.
  • I kindly ask if they have tested the change in sensitivity to other drugs for these first experiments.
  • In my opinion, this data have been evaluated not only on MDAMB231 cancer cells but might be interesting to know if the the impact towards a less aggressive or non-tumor cell line is the same to known if it is a specific response or not.
  • For the point “Alteration of cancer malignant phenotype by coculture with astrocytes” the authors to examine the changes on the astrocyte side, collected intracellular metabolites 48h after treatment with MDA231 culture medium, and then indirect coculture and single cultured astrocytes were compared. In my opinion the data at 24h and prolonged time if present should be reported or discuss this choice. 

Reviewer 3 Report

I have not competing interests to declare.

In this paper, the authors investigate the role of astrocytes in the interplay with metastatic brain tumors. In order to study the interactions between the two cell populations they took advantage of coculture experiments, using two different approaches. Given the central role of astrocytes in brain metabolism, authors analysed the metabolome of both cell types and highlighted differences in several amino acids pathways, in particular glutamate, when astrocytes and tumors cells were cultured together.

Some major points in the results section must be clarified and integrated:

  • Figures showing results are often presented in a too simplistic manner for example lacking of appropriate legends, axis title, treatment indications (i.e. Figure 1, panel B lacks of CDDP dosage)
  • It’s necessary to prove that eGFP emission is proportional to cell number in order to justify the use of this measure as a surrogate for cell viability. FACS analysis (PI staining) or MTT assay could be useful to have a more quantitative indication of vitality.
  • The method used to measure proliferation rate (namely dividing cell viability at 48h by that at 8h) is not the correct approach.
  • Number of proliferating cells followed astrocytes addition is defined “significant” however authors do not provide any statistical test to support this statement (see Figure 3, panel C).
  • Discussion needs to be rewritten in a more concise manner, with a specific focus on the results of the article and providing additional insights on the presented data.
  • The criteria that authors applied in the use of coculture, for example direct approach for migration phenotype analysis or indirect approach for metabolome analysis, need deeper explanation and justification.

Minor points:

  • Figure 1 legend: please add CDDP dosage in panel B.
  • Figure 2: please add axis title (panel B and C).
  • Figure 3: please change panel A group name accordingly to panel B and C. A statistical test is needed in panel 3.
  • Supplementary Figure 1 is not understandable.
  • Discussion, line 247 please modify with Fig.1B.
  • Discussion, lines 291-292 and lines 319-320: please check the repetition of the same sentence.

Round 2

Reviewer 1 Report

The authors have answered the reviewer's concerns.